# Establishment and Verification of the UAV Coupled Rotor Airflow Backward Tilt Angle Controller

**Han Wu, Dong Liu, Yinwei Zhao, Zongru Liu, Yunting Liang, Zhijie Liu, Taoran Huang** **, Ke Liang, Shaoqiang Xie and Jiyu Li \***

College of Engineering, South China Agricultural Universities, Guangzhou 510642, China; wuhan@scau.edu.cn (H.W.); liudong@stu.scau.edu.cn (D.L.); zhaoyinwei@stu.scau.edu.cn (Y.Z.); luzhi0439@stu.scau.edu.cn (Z.L.); daaize@stu.scau.edu.cn (Y.L.); liuzj@stu.scau.edu.cn (Z.L.); htr@stu.scau.edu.cn (T.H.); liangke1@stu.scau.edu.cn (K.L.); skmfmmy@stu.scau.edu.cn (S.X.)
* Correspondence: lijiyu@scau.edu.cn

**Abstract:** At present, all the flight controllers of agricultural UAVs cannot accurately and quickly control the influencing factors of the UAV coupled rotor airflow backward tilt angle during the application process. To solve the above problem, a Rotor Airflow Backward Tilt Angle (RABTA) controller is established in this paper. The RABTA controller integrates advanced sensor technology with a novel algorithmic approach, utilizing real-time data acquisition and state–space analysis to dynamically adjust the UAV's rotor airflow, ensuring precise control of the backward tilt angle. The control effect of the traditional flight controller and RABTA controller in the process of pesticide application and the corresponding operation effect are compared and analyzed. The comparison results show that the RABTA controller reduces the control error to less than 1 degree, achieving a 48.3% improvement in the uniformity of the distribution of pesticides droplets across the crop canopy, which means that the UAV field application effect is implemented and the innovation of the UAV field application control mode is realized.

**Keywords:** UAV; rotor airflow; flight controller; backward tilt angle



## 1. Introduction

In recent years, with the comprehensive development of agricultural aviation technology, rotor-wing UAVs have been widely used in field operations such as pesticide application and pollination because of their high field operation efficiency and low cost. With the increase in the field application of rotor-wing UAVs, the effect of droplet distribution uniformity and droplet drift rate in the application process has been widely studied. As a main factor that affects the field application of rotor-wing UAVs, rotor airflow has been extensively researched by scholars at home and abroad. Li et al. [1] combined the wind speed parameter acquisition system and the Beidou positioning system to fit the distribution law of rotor airflow on crop canopy planes during rotor-wing UAV operation. Li et al. [2] used a pitot tube sensor array to obtain vertical wind speed data at two specific heights within a rice canopy—30 cm and 60 cm beneath the canopy level (the position at a distance of 30 cm and 60 cm directly beneath the lowest point of the rice plant canopy). These measurements were crucial for analyzing the stratified distribution of rotor airflow both within the canopy and at the levels immediately below it, providing insights into how airflow dynamics change at different layers of the rice canopy. The above studies are based on the overall distribution law of rotor airflow, without refining the specific characteristics of rotor airflow and its influence on the field application effect.

With the increased research on rotor airflow, the specific characteristic parameter of the UAV coupled rotor airflow backward tilt angle (RABTA, hereafter referred to as "backward tilt angle") has become the focus of research. Wang et al. [3] used the lattice Boltzmann method (LBM) to simulate the rotor airflow of a six-rotor UAV under flight conditions

and studied the change rule of the backward tilt angle under different flight speeds, flight altitudes, crosswind speeds, workloads, and other characteristic parameters. Zhu et al. [4] adopted a three-dimensional computational fluid dynamics (CFD) model to establish the variation law of the backward tilt angle of a quadrotor UAV under different flight speeds (1–5 m/s). The above research shows that the backward tilt angle is the angle at which the rotor airflow generated by the rotor is tilted backward due to the influence of forward wind resistance and the environmental wind field during the flight of the rotor-wing UAV. During UAV operations, the accurate and rapid control of the backward tilt angle can improve the stability of rotor airflow in crop canopy coverage and reduce the influence of field interference factors on droplet drift to improve the effect of UAV field application. However, at present, all the flight controllers of agricultural UAVs do not take the backward tilt angle as the control object and cannot accurately and quickly control the influencing factors of the backward tilt angle during the application process.

Furthermore, there are no studies on agricultural UAV flight controllers involving factors that affect field operations, such as the backward tilt angle, at home and abroad, and most of the research on UAV flight controllers remains at the level of UAV safety performance. Liu et al. [5] designed the longitudinal attitude control system of agricultural UAVs by using the classical PID control method. Orsag, Poropat, and Bogdan [6] combined the classical PID control method with the LQR control method to form a hybrid control method for quad-rotor UAVs. Benallegue, Mokhtari, and Fridman [7] used a feedback linearization controller to control quadrotor UAVs. All these studies are based on linear control theory. The design of the control algorithm only considers the linearized mathematical model of the controlled object, ignoring the nonlinear part of the controlled object.

To supplement the nonlinear part of the control object in the control algorithm, many nonlinear control methods have been applied to the bottom control of quadrotor UAVs, and many research results have been achieved [8–11]. Raffo et al. [12] used integral prediction and H∞ nonlinear robust control methods to solve the path-tracking problem of quadrotor UAVs. Shi et al. [13] combined the backstepping control strategy with SMC to derive fractional backstepping SMC and obtained good control performance. Fujimoto et al. [14] simplified the small quadrotor UAV model and proposed a control method based on immersion and invariance (I&I) to overcome the uncertainty caused by the drag coefficient and thrust. The above studies all use nonlinear controllers to solve the low-level control problem of small UAVs. The disadvantage is that the controller depends on the precise mathematical model of the system. If the mathematical model is not sufficiently accurate, the stability of the system will be affected, and the system will even fail.

Because many nonlinear control algorithms depend on the exact mathematical model of the controlled object, the implementation of these algorithms is challenging. Therefore, many scholars began to study control methods without mathematical models, such as the active disturbance rejection control method [15–17], which uses an extended state observer (ESO) to estimate unknown models and disturbances and designs controllers based on the observer. However, many parameters in the extended state observer lack physical meaning and guidance, resulting in complicated parameter adjustment. In addition, the high-order differential feedback controller (HODFC) [18,19] can be used to solve the shortcomings of the above method by designing a high-order differential (HOD) observer to observe the input, output, and their derivatives and adopting differential feedback control and control flash. Although these control methods can make up for the problem that nonlinear control algorithms rely too much on precise mathematical models, they need to carry out complex parameters in the control process to ensure the compatibility of the controller to the control object and cannot achieve accurate and fast control of the backward tilt angle under the interference of unknown environmental wind fields.

In summary, all UAV flight controllers do not take the backward tilt angle as the control object and cannot accurately and quickly control the backward tilt angle, which is a factor that affects operations during pesticide application. To solve the above problems, a UAV coupled rotor airflow backward tilt angle controller (hereafter referred to as the

"RABTA controller") is established in Section 2. This controller is used to replace the traditional UAV flight controller to achieve accurate and rapid control of the backward tilt angle during UAV application. Moreover, in Section 3, the control effect of the RABTA controller and the corresponding operation effect of the traditional flight controller and RABTA controller in the application process of rotor airflow are compared and analyzed to prove the effectiveness of the RABTA controller established in this paper to solve the above problems and provide a new flight control mode for the field application of UAVs.

## 2. Materials and Methods

In this section, the establishment process of the RABTA controller is described. First, a set of rotor airflow backward tilt angle sensors (hereafter referred to as "RABTA sensors") is designed and produced in Section 2.1 to realize real-time state observation of the backward tilt angle during UAV flight, and the accuracy of the sensor observation data is verified through test experiments. Then, in Section 2.2, experiments of the backward tilt angle under different flight speeds are carried out, and state space equations of the backward tilt angle are constructed according to the data collected in the experiments. Finally, in Section 2.3, the RABTA controller is established according to the established state space equation in Section 2.2.

### 2.1. Design of the RABTA Sensor

#### 2.1.1. Principle of RABTA Sensor

As shown in Figure 1, during the flight of a rotor wing UAV, the rotor airflow generated by its rotorcraft is coupled with the forward wind resistance and the ambient wind field, forming a coupled rotor airflow with a backward tilt state. At this time, the RABTA sensor in the coupled rotor airflow is not affected by the forward wind resistance and ambient wind field due to the obstruction of the coupled rotor airflow and only wobbles under the combined wind force generated by the coupled rotor airflow and reaches a new force equilibrium position. The swing angle of the sensor in the equilibrium state is equal to the backward tilt angle. Therefore, the sensor swing angle is used to observe the backward tilt angle.

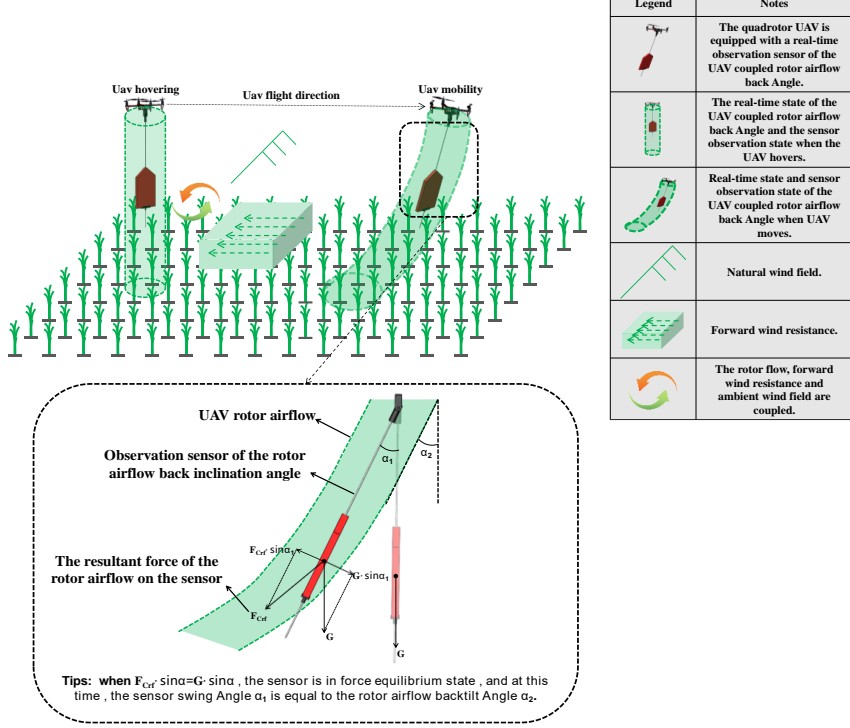

**Figure 1.** Schematic diagram of the principle of the RABTA sensor.

### 2.1.2. RABTA Sensor Structure

According to Section 2.1.1, the structural composition of the designed RABTA sensor is shown in Figure 2, and the specific parameters of each structure are shown in Table 1. To prevent the inertia of the sensor itself from affecting the observation effect of the sensor, a KT plate with low mass is selected as the material of the tail plate. In addition, to ensure that the tail plate is completely located inside the rotor airflow when the sensor is in balance so as not to be affected by the forward wind resistance and ambient wind field, the transverse width of the tail plate is designed to be equal to the distance between the adjacent rotors of the rotor wing UAV, and the distance between the top of the tail plate and the bottom of the UAV is designed to be half of the wheelbase of the rotor wing UAV. At this time, the length of the main straight rod is greater than the sum of the distance between the top of the tail plate and the bottom of the UAV and the longitudinal length of the tail plate to ensure that the landing wheel can be installed.

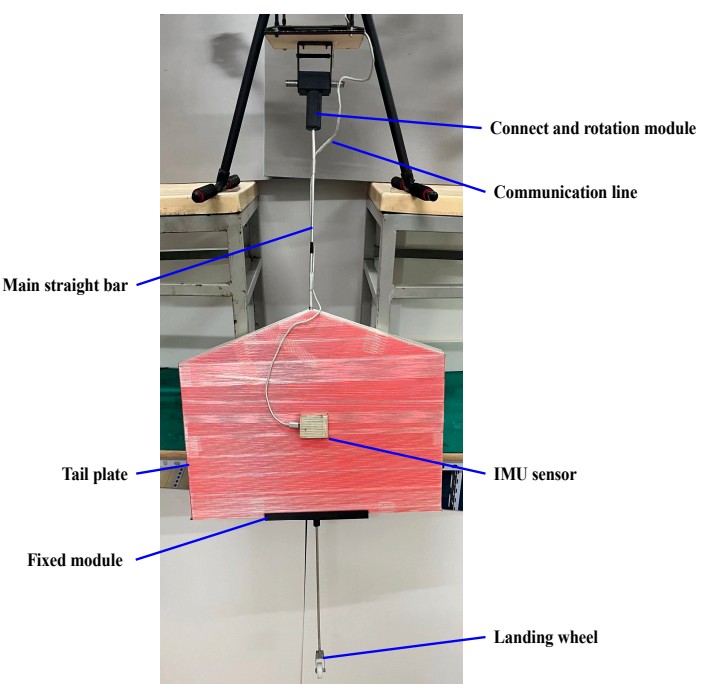

**Figure 2.** RABTA sensor physical image.

**Table 1.** Structure parameter table of RABTA sensor.

| Name | Material | Size (mm) | Weight (g) | Function |
|---|---|---|---|---|
| Connection and rotation module | PLA and aluminum | 114.5 × 56 × 25.5 | 31 | Used to connect UAVs and provide rotation function as a rotation axis |
| Communication line | Aluminum alloy and plastics | 1500 | 5 | Data communication |
| Main straight bar | Aluminum | Diameter: 5 Length: 100 | 11 | As the bearer of all modules |
| Tail plate | KT plate | 200 × 300 × 15 | 90.2 | Improve the wind sensing capability of the sensor |
| Fixed module | PLA | 25 × 250 × 30 | 3 | Fixed tail plate |
| IMU sensor | NA | 31.5 × 43.1 | 5.6 | Output the sensor swing Angle information |
| Landing wheel | Aluminum alloy and plastics | 40 × 18 × 36 | 4.2 | Enable the drone to land safely |

### 2.1.3. RABTA Sensor Observation Effect Verification Experiment

Test Site and Materials

The experimental site for the observation effect verification experiment of the RABTA sensor is the Innovation Laboratory of Engineering College of South China Agricultural University, Tianhe District, Guangzhou City, Guangdong Province (23°23′47.98″ N, 113°26′11.79″ E), and the experimental environment is indoors, without environmental wind field interference.

In this experiment, a BlueX450 quadrotor UAV (as shown in Figure 3a, manufactured by Shenzhen Yuangu Technology Co., Ltd. in Shenzhen, China) is selected to generate rotor airflow. The BlueX450 quadrotor UAV is firmly connected to the experimental test rack (as shown in Figure 3b). A RadioLink sAT9S remote control, which is manufactured by Shenzhen LeDi Electronics Co., Ltd. in Gurgaon, India, was used to drive the UAV motor to rotate at a fixed throttle value and maintain the speed. All materials used in the experiment are shown in Table 2.

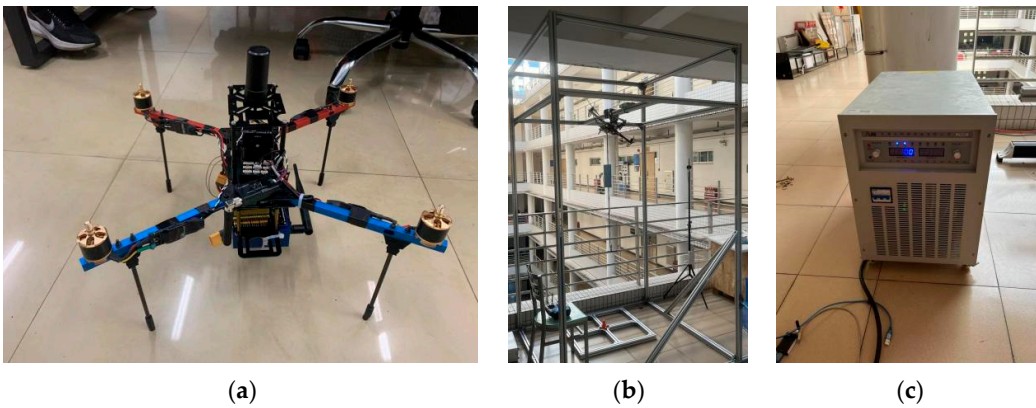

(a)      (b)      (c)

**Figure 3.** Physical drawings of the experimental materials: (**a**) BlueX450 quadrotor UAV; (**b**) the experimental test rack; (**c**) power supply.

**Table 2.** Detailed list of experimental materials.

| Device Name | Model Number | Main Features | Purpose |
|---|---|---|---|
| Quadrotor UAV | BlueX450 | Wheelbase: 450 mm<br>Type of motor: DJi 2216<br>Size of blade: 9 inches | Used to generate rotor airflow and its attitude is changed to simulate different backward tilt angle |
| Test frame | NA | 1.5 m × 1.5 m × 2.5 m | Fixed UAV |
| RABTA sensor | NA | As shown in Table 1 | It is used to realize the observation of backward tilt angle |
| Power supply | ANS 60V 300A Regulated voltage supply | 0.6 m × 0.6 m × 0.4 m | Powering drones |
| Remote control | RadioLink AT9S | NA | Controlling of the motor speed of the UAV |
| Laptop computer | Lenovo Savior Y520 | NA | Saving data |

Experimental Process and Result Analysis

As shown in Figure 4a, during the experiment, the motor speed of the BlueX450 UAV remained unchanged, and the attitude of the UAV was continuously increased from 0 to 40 degrees at an interval of 5 degrees. At the same time, the observation data of the RABTA sensor under different UAV attitudes are collected. The entire experiment is conducted in an indoor windless environment. At this time, the rotor airflow is perpendicular to the plane in which the motor of the BlueX450 UAV is located (as shown in Figure 4b), so the actual value of the backward tilt angle is equal to the attitude angle of the UAV. The actual value of the backward tilt angle is compared with the observed value of the RABTA sensor, and the results are shown in Figure 5.

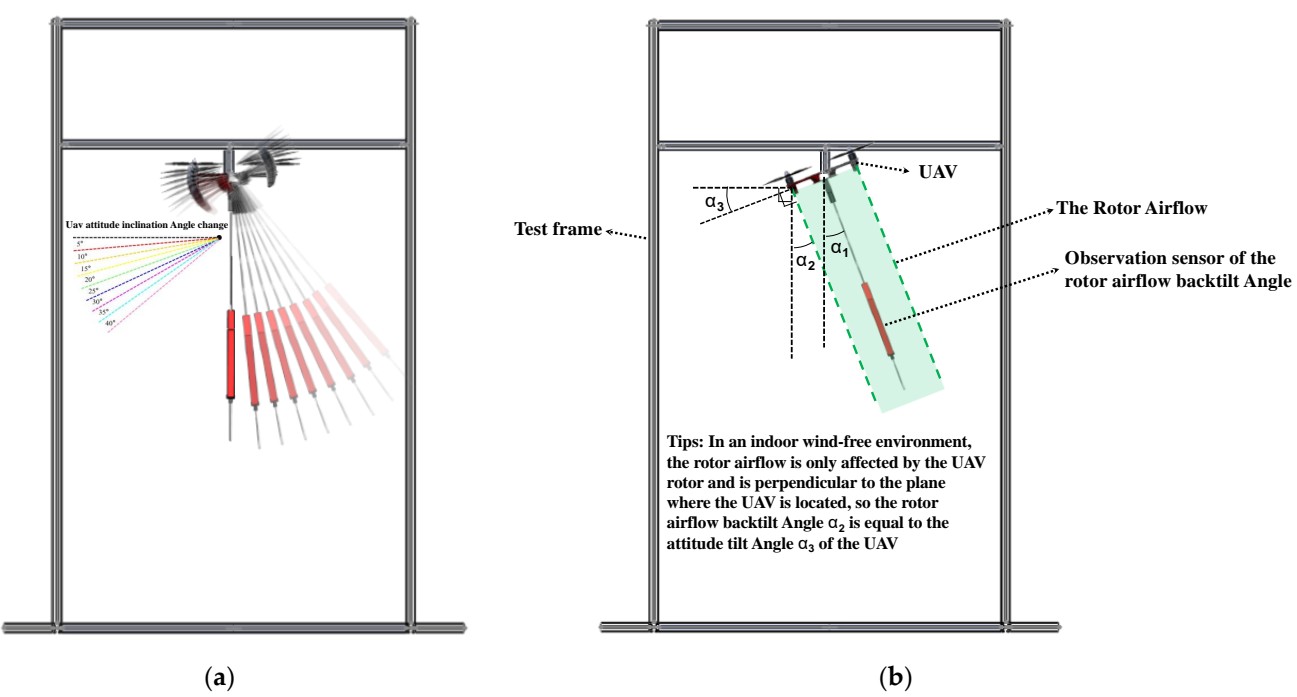

**Figure 4.** Experimental process diagram: (**a**) schematic diagram of the experimental process; (**b**) experimental parameters image resolution.

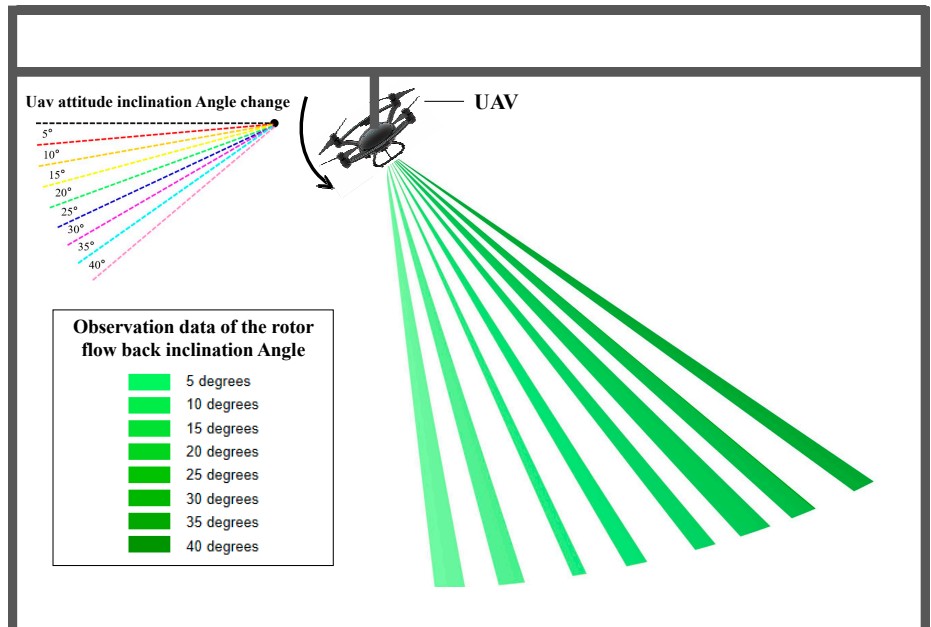

**Figure 5.** Experimental results.

During the experiment, the pitch attitude angle of the BlueX450 UAV gradually increased from 0 degrees to 40 degrees with an increment of 5 degrees. At this time, the simulated backward tilt angle also gradually increased from 0 degrees to 40 degrees with an increment of 5 degrees. Correspondingly, the observation results of the RABTA sensor also show a gradually increasing trend. In addition, as shown in Figure 6, under the actual state of different backward tilt angles, the observed data of the RABTA sensor have no abnormal values, indicating that the observed data are concentrated and have no obvious skewed distribution. Furthermore, the actual value of the backward tilt angle in each state is compared with the observed value of the RABTA sensor, as shown in Table 3.

Due to the inherent inertia of the RABTA sensor itself, the observed value of the backward tilt angle have a certain degree of error. The maximum average error of the observed value of the backward tilt angle is 0.52. The maximum error is 1.47. In addition, turbulence within the rotor airflow causes some fluctuations in the observed data of the RABTA sensor, but the fluctuation range of the observed data does not exceed 3. This shows that the RABTA sensor can accurately observe the real-time state of the backward tilt angle. In summary, the RABTA sensor can accurately and stably observe the state of the backward tilt angle.

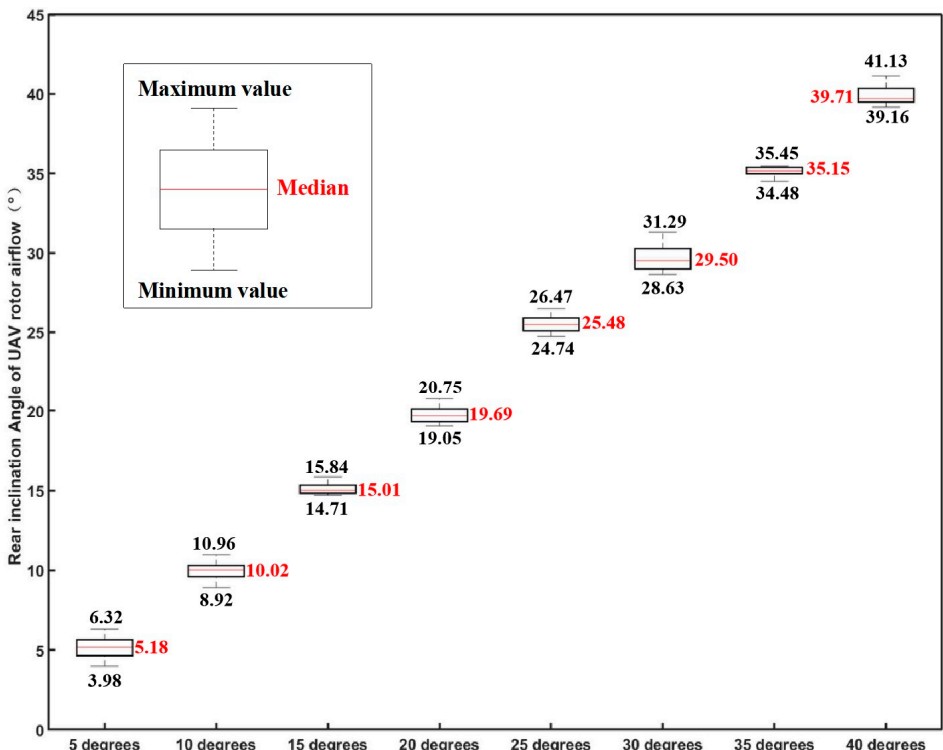

**Figure 6.** Box plot of the observation data analysis of the RABTA sensor.

**Table 3.** Comparison between the actual and observed values of RABTA sensor.

| Pitch Angle of the Rotor UAV (Degree) | Backward Tilt Angle (Degree) | Sensor Output Data (Degree) | | | Error (Degree) | | Range of Fluctuation (Degree) |
|---|---|---|---|---|---|---|---|
| | | Average Value | Maximum Value | Minimum Value | Average Error | Maximum Error | |
| 5 | 5 | 5.17 | 6.32 | 3.98 | 0.17 | 1.32 | 2.34 |
| 10 | 10 | 9.97 | 10.96 | 8.92 | −0.03 | 1.08 | 2.04 |
| 15 | 15 | 15.09 | 15.84 | 14.71 | 0.09 | 0.84 | 1.13 |
| 20 | 20 | 19.73 | 20.75 | 19.05 | −0.27 | 0.95 | 1.70 |
| 25 | 25 | 25.52 | 26.47 | 24.74 | 0.52 | 1.47 | 1.73 |
| 30 | 30 | 29.65 | 31.29 | 28.63 | −0.35 | 1.37 | 2.66 |
| 35 | 35 | 35.14 | 35.45 | 34.48 | 0.14 | 0.52 | 0.96 |
| 40 | 40 | 39.93 | 41.13 | 39.16 | −0.07 | 1.13 | 1.98 |

*2.2. Collection Experiment of the Backward Tilt Angle Observation State at Different Flight Speeds*

2.2.1. Test Site and Materials

The experimental location of the backward tilt angle observation at different flight speeds is Huashan Stadium, South China Agricultural University, Tianhe District, Guangzhou, Guangdong Province (23°23′47.98″ N, 113°26′11.79″ E). Since the total mass of the airborne equipment in the experiment exceeds 1.5 kg and the flight duration exceeds 20 min, the ZD680 quadrotor UAV (as shown in Figure 7, manufactured by Dongguan Volcano Model

Technology Co., Ltd. in Dongguan, China) is selected to perform flight tasks at different speeds in this experiment, and the RABTA sensor is used to collect the observation state of the backward tilt angle. All materials used in the experiment are shown in Table 4.

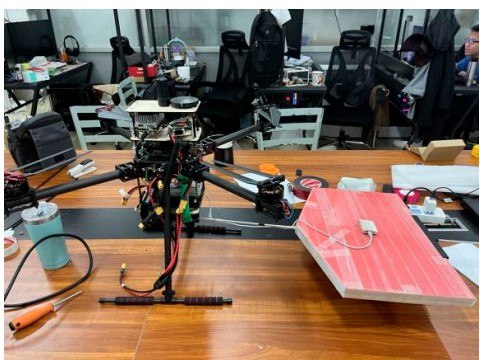 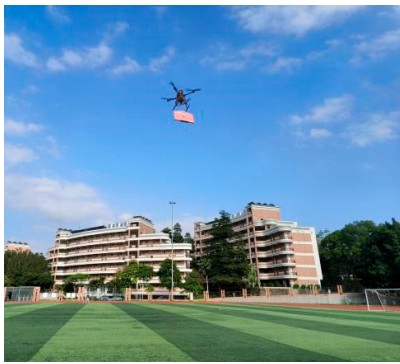

**Figure 7.** Drawings of the UAV and other materials used in the experiment.

**Table 4.** Detailed list of experimental materials.

| Material Name | Quantity | Main Features | Uses |
|---|---|---|---|
| ZD680 quadrotor UAV | 1 | Wheelbase: 680 mm<br>Type of motor: SUNNYSKY X4110S<br>Size of blade: 15 inches<br>Type of battery: 6S 10,000 mAh<br>Type of Flight control: Pixhawk cube black<br>Rated load: Over 5 kg<br>Endurance time: Over 20 min | It is used to carry out all the flights required for the experiment. |
| Beidou RTK positioning System | 1 | Accuracy of positioning: Centimeter level<br>Type of output: NMEA<br>Quality: 14 g | It is used to provide centimeter-level position data and speed data for UAVs. |
| RABTA sensor | 1 | Connecting rod length: 100 cm<br>Wind plate shape: Tail airfoil type<br>Size of air sensing plate: 48 cm × 36 cm<br>Quality of sensor: 320 g | It is used to obtain the observed value of the backward tilt angle during the actual flight of the UAV. |
| Jetson TX2 airborne processor | 1 | Quality: 251 g | It is used to save the observed values of the UAV speed and backward tilt angle. |

### 2.2.2. Experimental Methods and Parameters

During the experiment, the ZD680 quadrotor UAV carrying the RABTA sensor performs the fixed speed linear flight task, as shown in Figure 8. The ZD680 quadrotor UAV takes off from the take-off point to a height of 5 m and then lands at the landing point after straight flight in the acceleration phase, constant speed phase, and deceleration phase. During the flight, to prevent the influence of the ground effect of the rotor airflow on the experimental results, the height of the UAV is always maintained at 5 m, and the farthest linear flight path (45 m distance) is selected under the limitation of the experimental site. There is no environmental wind interference throughout the flight. In addition, considering the speed limit of the quadrotor UAV in a field operation, the maximum flight speed of this experiment is 4 m/s, and the linear route task of eight sorties is set at the level of 0.5 m/s in the speed interval of 0 m/s to 4 m/s. The flight range, heading, flight altitude, and ambient wind field of the eight sorties were maintained consistent to ensure that only one variable, the flight speed, existed in the flight process of each sortie. The corresponding relationship between the above two variables is shown in Table 5.

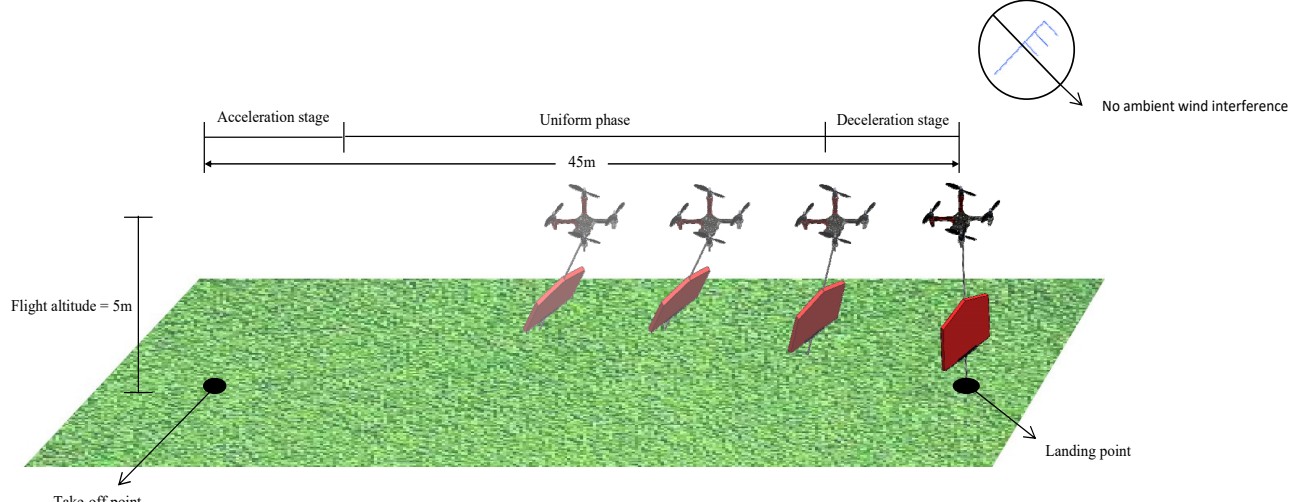

**Figure 8.** Schematic diagram of the constant speed straight flight mission.

**Table 5.** Flight data sheets for eight sorties.

| Flight Sorties | Flight Speed (m/s) | Height over Terrain (m) | Flight Range(m) | | | Collect Data |
|---|---|---|---|---|---|---|
| | | | Accelerated Range | Constant Range | Retarded Range | |
| 1 | 0.5 | 5 | 0.6 | 43.6 | 0.8 | |
| 2 | 1 | 5 | 1.9 | 41.8 | 1.3 | |
| 3 | 1.5 | 5 | 2.0 | 40.8 | 2.1 | Flight speed and observation state of backward tilt angle at constant-velocity stage |
| 4 | 2 | 5 | 2.3 | 40.1 | 2.6 | |
| 5 | 2.5 | 5 | 3.4 | 38.2 | 3.3 | |
| 6 | 3 | 5 | 4.4 | 36.8 | 3.8 | |
| 7 | 3.5 | 5 | 5.2 | 34.4 | 5.4 | |
| 8 | 4 | 5 | 6.0 | 32.3 | 6.7 | |

### 2.2.3. Experimental Results

As shown in Figure 9, with the gradual increase in the UAV flight speed, the observation state of the backward tilt angle also shows an increasing trend. The corresponding relationship between the above two is shown in Equation (1).

$$\alpha = -0.2811v^5 + 2.5895v^4 - 7.872v^3 + 8.782v^2 + 6.2417v + 0.0287, \tag{1}$$

where $\alpha$ is the observation state of the backward tilt angle and $v$ is the flight speed of the UAV. According to Formula (1), the corresponding data of the UAV flight speed and the observation state of the backward tilt angle that satisfy the corresponding relationship are extracted, as shown in Table 6.

The MATLAB system identification tool is used to identify the control system data in Table 6, and the identified control transfer function between the backward tilt angle and the flight speed of the UAV is shown as follows:

$$G(s) = \frac{\alpha(s)}{V(s)} = \frac{21.17}{s + 2.252}, \tag{2}$$

where $\alpha(s)$ is the real-time state of the backward tilt angle and $V(s)$ is the flight speed of the UAV. Based on Equation (2), the integral control relationship between the UAV flight speed and flight acceleration is added to obtain the control transfer function between the

backward tilt angle and flight acceleration, and it is converted into the state space equation shown in Equation (3).

$$\begin{cases} \dot{x}(t) = \begin{bmatrix} 0 & 1 \\ 0 & -2.252 \end{bmatrix} * x(t) + \begin{bmatrix} 0 \\ 21.17 \end{bmatrix} * v(t) \\ y(t) = \begin{bmatrix} 1 & 0 \end{bmatrix} * x(t) \end{cases} \tag{3}$$

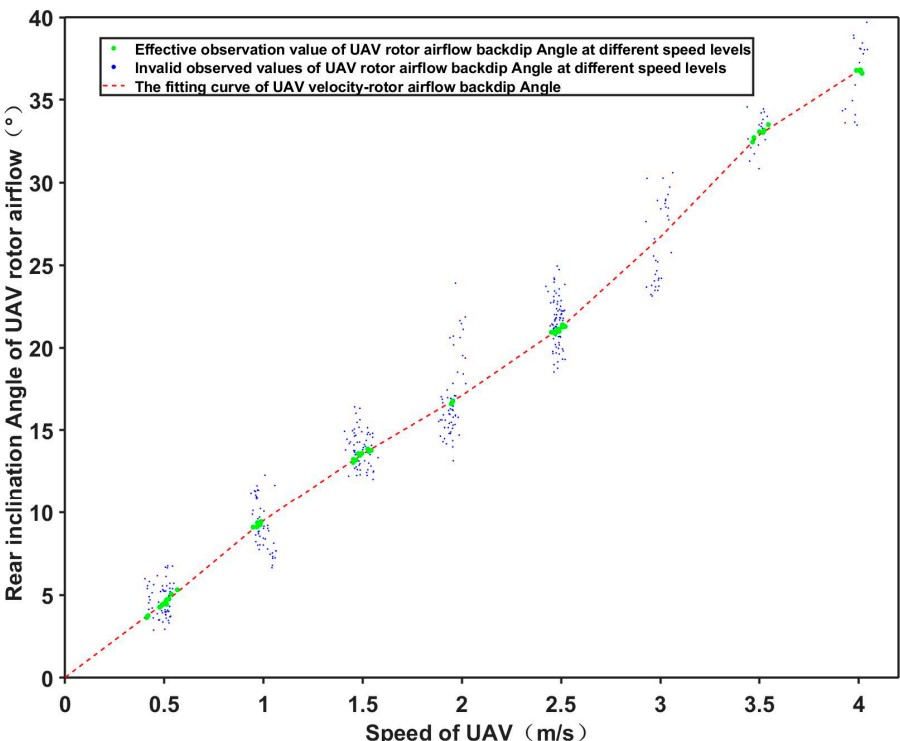

**Figure 9.** Experimental results between the rear angle of inclination of the airflow of the UAV rotor and the speed of the UAV.

**Table 6.** The UAV flight speed corresponds to the list of backward tilt angles.

| v (m/s) | α (Degree) | v (m/s) | α (Degree) | v (m/s) | α (Degree) | v (m/s) | α (Degree) |
|---|---|---|---|---|---|---|---|
| 0.41 | 3.64 | 0.95 | 9.08 | 1.53 | 13.71 | 2.52 | 21.27 |
| 0.42 | 3.75 | 0.97 | 9.09 | 1.54 | 13.78 | 3.46 | 32.44 |
| 0.48 | 4.26 | 0.97 | 9.35 | 1.95 | 16.59 | 3.47 | 32.71 |
| 0.49 | 4.38 | 0.98 | 9.22 | 1.95 | 16.77 | 3.5 | 33.07 |
| 0.49 | 4.42 | 0.99 | 9.42 | 2.45 | 20.93 | 3.52 | 33.03 |
| 0.51 | 4.60 | 1.45 | 13.05 | 2.46 | 20.94 | 3.52 | 33.15 |
| 0.51 | 4.58 | 1.45 | 13.23 | 2.47 | 20.85 | 3.54 | 33.5 |
| 0.51 | 4.44 | 1.47 | 13.2 | 2.48 | 21.11 | 3.99 | 36.78 |
| 0.51 | 4.70 | 1.48 | 13.59 | 2.49 | 21.09 | 4.01 | 36.74 |
| 0.52 | 4.76 | 1.49 | 13.49 | 2.49 | 20.99 | 4.01 | 36.81 |
| 0.53 | 5.04 | 1.50 | 13.6 | 2.51 | 21.39 | 4.02 | 36.61 |
| 0.57 | 5.30 | 1.52 | 13.86 | 2.51 | 21.25 | | |

### 2.3. Establish RABTA Controller

According to the state space equation of the backward tilt angle constructed in Section 2.2, the structure diagram of the RABTA controller is designed as shown in Figure 10. The differential tracker (DT) and the expanded state observer (ESO) are used to improve the control accuracy of the controller, and the *fal* nonlinear controller is used to shorten

the regulation process of the controller. In addition, two modules, an integral converter and a position discriminator, are added to the controller to improve the compatibility of the controller for traditional UAV flight so that it can replace the traditional UAV flight controller and take the backward tilt angle as the control object in the actual field operation of the UAV to realize the accurate control and fast control of the backward tilt angle during the operation process.

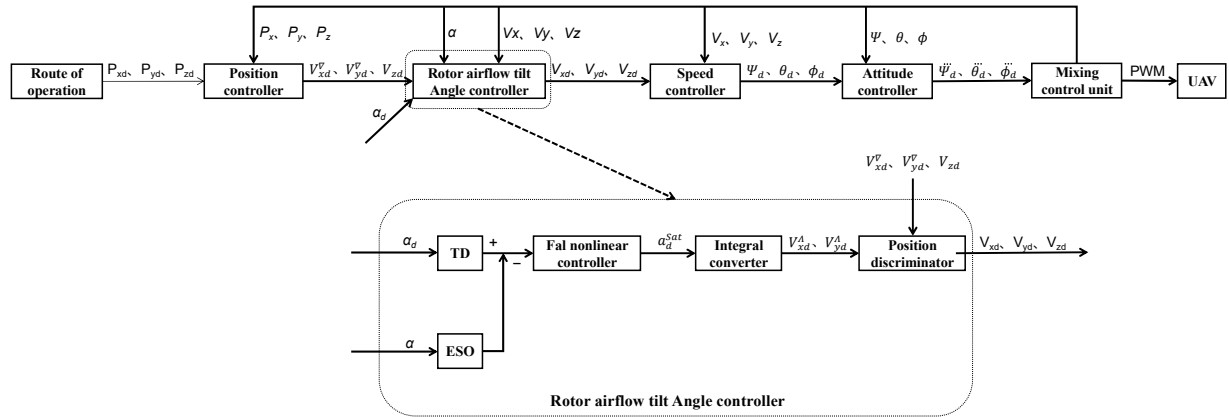

**Figure 10.** The structure diagram of RABTA controller.

### 2.3.1. The Differential Tracker

The differential tracker of the ADRC control algorithm is used in the RABTA controller developed in this paper. Its function is to obtain the expected angle and the expected angular velocity of the backward tilt angle. The expected angle of the backward tilt angle is converted into the expected angle and the expected angular velocity through the optimal synthesis function (fhan). The algorithm for the differential tracker is expressed as follows:

$$\begin{cases} \alpha_{1d}(k+1) = \alpha_{1d}(k) + h \cdot \alpha_{2d}(k) \\ \alpha_{1d}(k+1) = \alpha_{2d}(k) + h \cdot fst[\alpha_{1d}(k) - \alpha_d, \alpha_{2d}(k), r, h_0] \\ fst = fhan[\alpha_{1d}(k) - \alpha_d, \alpha_{2d}(k), r, h_0] \\ \quad = -r \cdot \frac{a}{2d} \cdot [sign(a+d) - sign(a-d)] - r \cdot sign(a) \cdot \left\{ 1 - \frac{1}{2} \cdot [sign(a+d) - sign(a-d)] \right\} \\ a = \frac{1}{2}(a_0 + y) \cdot [sign(y+d) - sign(y-d)] + a_2 \cdot \left\{ 1 - \frac{1}{2} \cdot [sign(y+d) - sign(y-d)] \right\} \\ a_2 = a_0 + \frac{1}{2} sign(y) \cdot (a_1 - d) \\ a_1 = \sqrt{d \cdot (d + 8|y|)} \\ y = \alpha_{1d}(k) - \alpha_d + a_0 \\ a_0 = h_0 \cdot \alpha_{2d}(k) \\ d = r \cdot h_0^2, \end{cases}$$

(4)

where $\alpha_{1d}(k+1)$ and $\alpha_{2d}(k+1)$ are the expected angle and the expected angular velocity of the backward tilt angle at the next moment, respectively, and $\alpha_{1d}(k)$ and $\alpha_{2d}(k)$ are the expected angle and the expected angular velocity of the backward tilt angle at the current moment, respectively. $h$ is the control step. $fst$ is the optimal synthesis function, whose input parameters are the difference value between the expected angle of the backward tilt angle at the current moment and the expected angle of the final backward tilt angle ($\alpha_{1d}(k) - \alpha_d$), the expected angular velocity of the backward tilt angle at the current moment ($\alpha_{2d}(k)$), the filtering factor ($r$), and the control step size ($h_0$).

### 2.3.2. Expanded State Observer (ESO)

In this paper, the expanded state observer is established according to the state space equation of the backward tilt angle constructed in Section 2.2. Its function is to expand the real-time angle of the backward tilt angle observed by the RABTA sensor into two

observation states of backward tilt angle and angular velocity. The algorithm of the state expansion observer is expressed as follows:

$$
\begin{cases}
e = \alpha_1(k) - y(k) \\
\alpha_1(k+1) = y(k) + \acute{\alpha}_1(k) \\
\alpha_2(k+1) = \alpha_2(k) + \acute{\alpha}_2(k) - h \cdot \beta \cdot fal(e, \vartheta, \delta) \\
fal(e, \vartheta, \delta) = \begin{cases} \frac{e}{\vartheta^{\delta-1}}, & |e| \leq \delta \\ |e|^{\vartheta} \cdot sign(e), & |e| > \delta \end{cases}
\end{cases}
, \tag{5}
$$

where $y(k)$ is the current backward tilt angle observed by the real-time state observation sensor, $\alpha_1(k)$ is the current backward tilt angle output by the expanded state observer, and $\alpha_2(k)$ is the current angular velocity of the tilt angle of the rotor flow output by the expanded state observer. $\alpha_1(k+1)$ and $\alpha_2(k+1)$ are the backward tilt angle state and angular velocity state of the rotor airflow at the next moment output by the expanded state observer, respectively. $\acute{\alpha}_1(k)$ and $\acute{\alpha}_2(k)$ are the angle increment and angular velocity increment of the rotor airflow at the current time, respectively, and the evaluation process is shown as follows:

$$
\begin{bmatrix} \acute{\alpha}_1(k) \\ \acute{\alpha}_2(k) \end{bmatrix} = \begin{bmatrix} 0 & 1 \\ 0 & -2.252 \end{bmatrix} \cdot \begin{bmatrix} \alpha_1(k) \\ \alpha_2(k) \end{bmatrix} + \begin{bmatrix} 0 \\ 21.17 \end{bmatrix} \cdot a_d(k), \tag{6}
$$

where $a_d(k)$ is the expected acceleration of the UAV output by the RABTA controller.

### 2.3.3. *fal* Nonlinear Controller

Considering the problem of the control range of the rotor airflow angle, the *fal* nonlinear controller is proposed as the control unit of the rotor airflow angle control. The error feedback is obtained according to the expected angle and expected angular velocity of the rotor airflow backtilt angle obtained in Section 2.3.1 and the observation state of the rotor airflow backtilt angle and angular velocity obtained in Section 2.3.2. At the same time, the expected acceleration of the UAV (based on the body coordinate system) is calculated by using the nonlinear function. The algorithm of the nonlinear controller is expressed as follows:

$$
\begin{cases}
e_1 = \alpha_{1d}(k) - \alpha_1(k) \\
e_2 = \alpha_{2d}(k) - \alpha_2(k) \\
a_d(k) = \beta_1 \cdot fal(e_1, \vartheta_1, \delta) + \beta_2 \cdot fal(e_2, \vartheta_2, \delta)
\end{cases}
. \tag{7}
$$

In addition, a saturation function is added to improve the control performance of the UAV. The expression for the saturation function is as follows:

$$
a_d^{Sat} = Sat(a_d) = \begin{cases}
a_{d_{max}}, & if\ a_d \geq a_{d_{max}} \\
a_d, & if\ a_{d_{min}} < a_d < a_{d_{max}} \\
a_{d_{min}}, & if\ a_d \leq a_{d_{min}}
\end{cases}
, \tag{8}
$$

where $a_{d_{max}}$ is the maximum acceleration of the UAV and $a_{d_{min}}$ is the minimum acceleration of the UAV. In summary, the fal nonlinear controller can be expressed as follows:

$$
a_d^{sat}(k) = Sat(\beta_1 fal(\alpha_{1d}(k) - \alpha_1(k), \partial_1, \delta) + \beta_2 fal(\alpha_{2d}(k) - \alpha_2(k), \partial_2, \delta)). \tag{9}
$$

### 2.3.4. Integral Converter and Position Discriminator

To solve the application problem of the above fal nonlinear controller in the UAV control system, an integral converter module is added to the RABTA controller, which is used to realize the integral conversion and coordinate system rotation between the expected acceleration of the UAV output by the fal nonlinear controller and the expected velocity of the UAV. The expression for the integral converter is as follows:

$$
v_{xd}^{\Lambda}(k+1) = v_{xd}^{\Lambda}(k) + \int a_d^{sat}(k) \cdot cos(\psi) dt, \tag{10}
$$

$$v_{yd}^{\Lambda}(k+1) = v_{yd}^{\Lambda}(k) + \int a_d^{sat}(k) \cdot sin(\psi) dt, \tag{11}$$

where $v_{xd}^{\Lambda}(k+1)$ and $v_{yd}^{\Lambda}(k+1)$ are the horizontal expected velocity components of the UAV at the next moment output by the RABTA controller. $v_{xd}^{\Lambda}(k)$ and $v_{yd}^{\Lambda}(k)$ are the horizontal expected velocity components of the UAV at the current moment output by the RABTA controller. In addition, to apply the RABTA controller to the straight flight route task, the position discriminator module is added to the end of the whole controller. The position discriminator module can select the input source of the expected speed according to the distance between the UAV and the expected waypoint to realize the straight flight path of the UAV. The expression of the position discriminator is shown in Equation (12), where $v_{xd}^{\nabla}$ and $v_{yd}^{\nabla}$ are the horizontal expected velocity components output by the UAV position controller.

$$\begin{bmatrix} v_{xd} \\ v_{yd} \end{bmatrix} = \begin{cases} \begin{bmatrix} 0 & 0 & 1 & 0 \\ 0 & 0 & 0 & 1 \end{bmatrix} \cdot \begin{bmatrix} v_{xd}^{\nabla} \\ v_{yd}^{\nabla} \\ v_{xd}^{\Lambda} \\ v_{yd}^{\Lambda} \end{bmatrix} \\ \\ \begin{bmatrix} 1 & 0 & 0 & 0 \\ 0 & 1 & 0 & 0 \end{bmatrix} \cdot \begin{bmatrix} v_{xd}^{\nabla} \\ v_{yd}^{\nabla} \\ v_{xd}^{\Lambda} \\ v_{yd}^{\Lambda} \end{bmatrix} \end{cases} \tag{12}$$

## 3. Experimental Results and Discussion

To verify the control effect and operation effect of the RABTA controller in the actual application process, a ZD680 quadrotor UAV is used in the real environment to carry out actual flight test experiments under different control modes and different expected angles of the backward tilt angle. At the same time, the difference in the control effect of the rotor airflow back inclination angle between the traditional flight controller and the RABTA controller and its influence on the actual operation effect of the UAV in the field are discussed.

### 3.1. Evaluation Metrics

To intuitively analyze the control effect of the RABTA controller during the operation and its influence on the actual field operation effect of the UAV, the step response dynamic performance index of each controller on the backward tilt angle control is used in the actual flight process as the evaluation index of the control effect of the RABTA controller. At the same time, in this paper, an ideal rotor airflow model in the process of UAV spraying flight operation is constructed (as shown in Figure 11), and the coverage area of the ideal rotor airflow generated by the UAV at different flight heights on the crop crown level (*S1*) and the droplet drift rate ($\tau$) of the UAV at different flight heights are used as the evaluation indices of the UAV field operation effect.

The ideal rotor airflow model shown in Figure 11 features a central axis and circular cross-sections with radii varying perpendicularly from the axis. The central axis ($Z = -100/X$, $X < 0$) has a fixed length, indicating airflow dissipation after a constant distance. The initial section radius is 0.34 m, with larger radii for further sections based on their distance from the starting point.

Coverage area (*S1*) within the crop canopy is the rotor airflow's overlap with the canopy plane. Close to the canopy, the airflow's entire cutoff section is below, affecting the canopy fully and forming an oval area. Farther away, the cutoff intersects the canopy, creating a partial elliptical area. Above the canopy, the airflow's upper section is the drifting region, while the lower section is the non-drifting region, with their ratio determining the droplet drift rate ($\tau$).

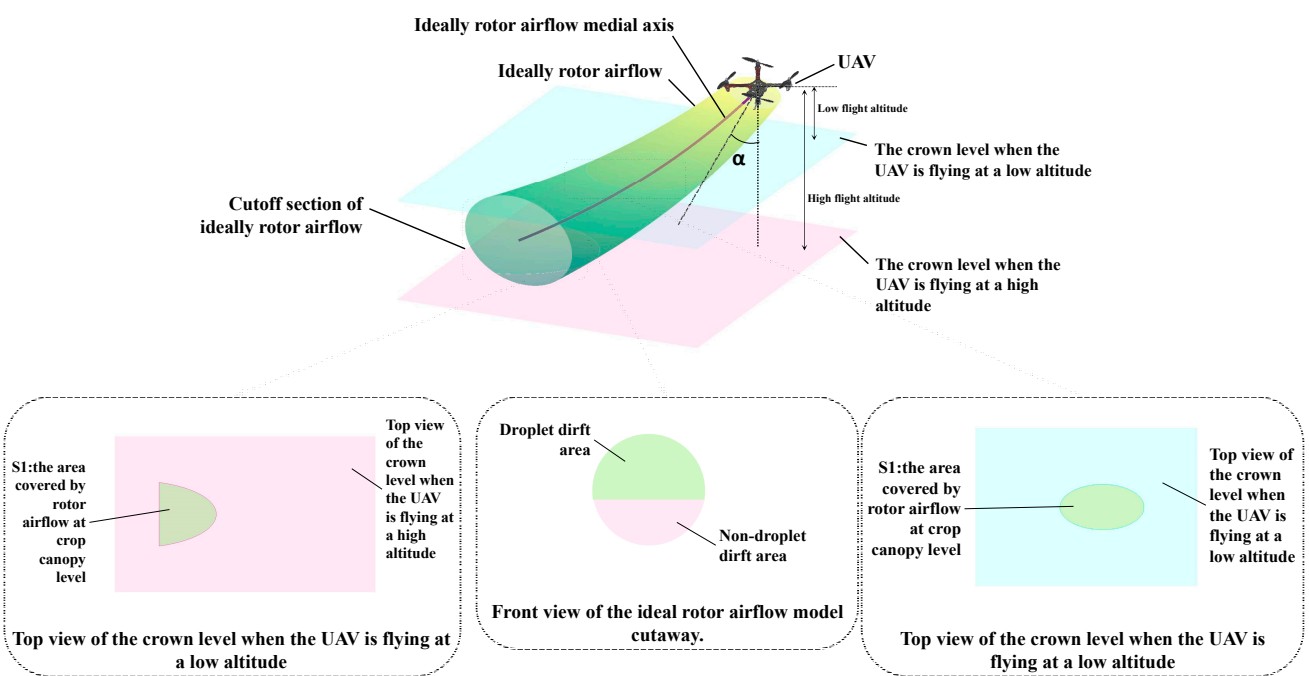

**Figure 11.** Schematic diagram of the ideal rotor airflow model.

Among them, the coverage area (*S1*) and the droplet drift rate ($\tau$) are related to the height of the UAV from the crop canopy ($H_{UAV\_to\_Canopy}$) and backward tilt angle ($\alpha$). The specific functional relationship is shown in Equation (13), where $CX_{start}$ is the x-coordinate of the starting point of the central axis of the rotor airflow and $RZ$ is the spatial z-coordinate of the crop canopy level. $CX_{max}$ and $CX_{min}$ are the x-coordinates of the vertex closest to the UAV and the vertex farthest away from the UAV in the coverage of the rotor airflow on the crop canopy, respectively. The evaluation and solution formulae are shown in Equations (14) and (15).

$$
\begin{cases}
S_1 = 2 * \int_{CX_{min}}^{CX_{max}} sqrt\left[r^2 - (1+k^2) * \left(RZ + \frac{100}{CX}\right)^2\right] * \left[CX - \left(RZ + \frac{100}{CX}\right) * k\right] d_{cx} \\
r = 0.34 + \left[ZL_{start} - CX * hypergeom\left(\left[-\frac{1}{2}, -\frac{1}{4}\right], \frac{3}{4}, -\frac{1}{CX^4}\right)\right] \\
ZL_{start} = CX_{start} * hypergeom\left(\left[-\frac{1}{2}, -\frac{1}{4}\right], \frac{3}{4}, -\frac{1}{CX_{start}^4}\right) \\
RZ = -\frac{100}{CX_{start}} - H_{UAV\_to\_Canopy} \\
CX_{start} = -sqrt\left(\frac{100}{tan\left(\frac{\pi}{2} - \alpha/180*3.14\right)}\right)
\end{cases}
\tag{13}
$$

$$
\begin{cases}
\left[\left(\frac{100}{CX_{min}^2}\right)^2 + 1\right] * \left(RZ + \frac{100}{CX_{min}}\right)^2 = r_{max}^2 \\
r_{max} = 0.34 + 0.1 * [ZL_{start} - ZL_{min}] \\
RZ = -\frac{100}{CX_{start}} - H_{UAV\_to\_Canopy} \\
ZL_{start} - ZL_{end} = 5 \\
ZL_{start} = CX_{start} * hypergeom\left(\left[-\frac{1}{2}, -\frac{1}{4}\right], \frac{3}{4}, -\frac{1}{CX_{start}^4}\right) \\
ZL_{end} = CX_{end} * hypergeom\left(\left[-\frac{1}{2}, -\frac{1}{4}\right], \frac{3}{4}, -\frac{1}{CX_{end}^4}\right) \\
ZL_{min} = CX_{min} * hypergeom\left(\left[-\frac{1}{2}, -\frac{1}{4}\right], \frac{3}{4}, -\frac{1}{CX_{min}^4}\right) \\
CX_{start} = -sqrt\left(\frac{100}{tan\left(\frac{\pi}{2} - \alpha/180*3.14\right)}\right) \\
CX_{end} < CX_{min} < -\frac{100}{RZ}
\end{cases}
\tag{14}
$$

$$\begin{cases} \left[\left(\frac{100}{CX_{max}^2}\right)^2 + 1\right] * \left(RZ + \frac{100}{CX_{max}}\right)^2 = r_{min}^2 \\ r_{min} = 0.34 + 0.1 * [ZL_{start} - ZL_{max}] \\ ZL_{start} = CX_{start} * hypergeom\left(\left[-\frac{1}{2}, -\frac{1}{4}\right], \frac{3}{4}, -\frac{1}{CX_{start}^4}\right) \\ ZL_{max} = CX_{max} * hypergeom\left(\left[-\frac{1}{2}, -\frac{1}{4}\right], \frac{3}{4}, -\frac{1}{CX_{max}^4}\right) \\ RZ = -\frac{100}{CX_{start}} - H_{UAV\_to\_Canopy} \\ -\frac{100}{RZ} < CX_{min} < CX_{start} \end{cases} \tag{15}$$

In addition, the drift rate of fog droplets ($\tau$) is also related to the height of the UAV from the crop canopy ($H_{UAV\_to\_Canopy}$) and backward tilt angle ($\alpha$). The specific functional relationship is shown in Equation (16). In this equation, $CX_{start}$ is the x-coordinate of the starting point of the central axis of the rotor airflow. $[RZ\ RX\ RY]$ is the vertex space coordinate of the boundary between the cutoff section of the rotor air flow and the crop crown level. $CX_{end}$ is the x-coordinate of the end point of the central axis of the rotor airflow, and its value can be calculated by Equation (17).

$$\begin{cases} \tau = \left[1 - \frac{r_{end}^2 * arcsin\left(\frac{abs(RY)}{r_{end}}\right) - sqrt\left(r_{end}^2 - abs(RY)^2\right) * abs(RY)}{\pi * r_{end}^2}\right] * 100\% \\ r_{end} = 0.34 + [ZL_{start} - ZL_{end}] \\ ZL_{start} = CX_{start} * hypergeom\left(\left[-\frac{1}{2}, -\frac{1}{4}\right], \frac{3}{4}, -\frac{1}{CX_{start}^4}\right) \\ ZL_{end} = CX_{end} * hypergeom\left(\left[-\frac{1}{2}, -\frac{1}{4}\right], \frac{3}{4}, -\frac{1}{CX_{end}^4}\right) \\ RY = sqrt\left[r_{end}^2 - (RX - CX_{end})^2 - \left(RZ + \frac{100}{CX_{end}}\right)^2\right] \\ RX = CX_{end} - \left(RZ + \frac{100}{CX_{end}}\right) * \frac{100}{CX_{end}^2} \\ RZ = -\frac{100}{CX_{start}} - H_{UAV\_to\_Canopy} \\ CX_{start} = -sqrt\left[\frac{100}{tan\left(\frac{\pi}{2} - \alpha/180*3.14\right)}\right] \end{cases} \tag{16}$$

$$\begin{cases} ZL_{start} - ZL_{end} = 5 \\ ZL_{start} = CX_{start} * hypergeom\left(\left[-\frac{1}{2}, -\frac{1}{4}\right], \frac{3}{4}, -\frac{1}{CX_{start}^4}\right) \\ ZL_{end} = CX_{end} * hypergeom\left(\left[-\frac{1}{2}, -\frac{1}{4}\right], \frac{3}{4}, -\frac{1}{CX_{end}^4}\right) \end{cases} \tag{17}$$

*3.2. Experimental Results*

During the experiment, the ZD680 quadrotor UAV carried out the flight of a 50 m straight-line mission in three control modes: position control, speed control, and backward tilt angle control. Among them, position control and speed control were carried out for one flight each, and backward tilt angle control was carried out for three flights. The expected angles of the rotor airflow are 20 degrees, 30 degrees, and 40 degrees. The flight range, heading, take-off point, and landing point of the five sorties are all the same. During the flight, the RABTA sensor is used to collect the real-time angle of the backward tilt angle and save it in line with the flight range and flight time of the UAV. The corresponding relationship between the flight range of the UAV and the real-time angle of the backward tilt angles of the five sorties is obtained, as shown in Figure 12. According to Equations (13) and (16), the corresponding relationship between the UAV flight range of five sorties and the coverage area of the rotor airflow on the crop crown level, as well as the corresponding relationship between the UAV flight range and the drift rate of fog droplets under the control the of backward tilt angle, are shown in Figures 13 and 14, respectively.

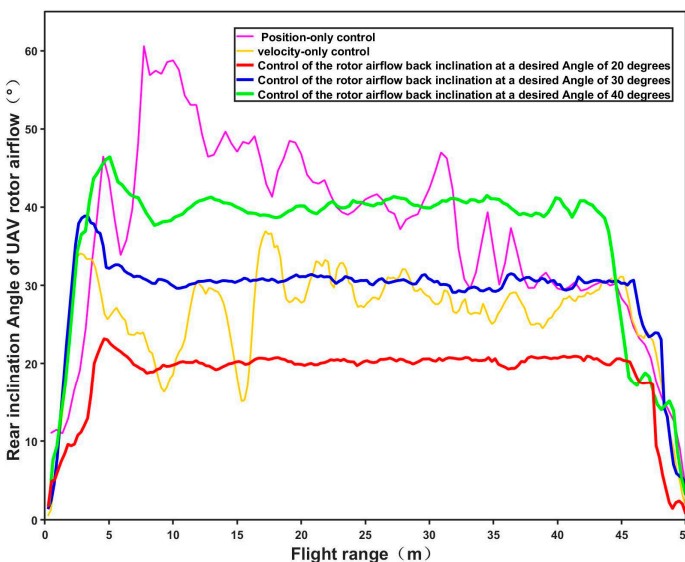

**Figure 12.** Experimental results for rear inclination with flight range at different designed angles and different control modes.

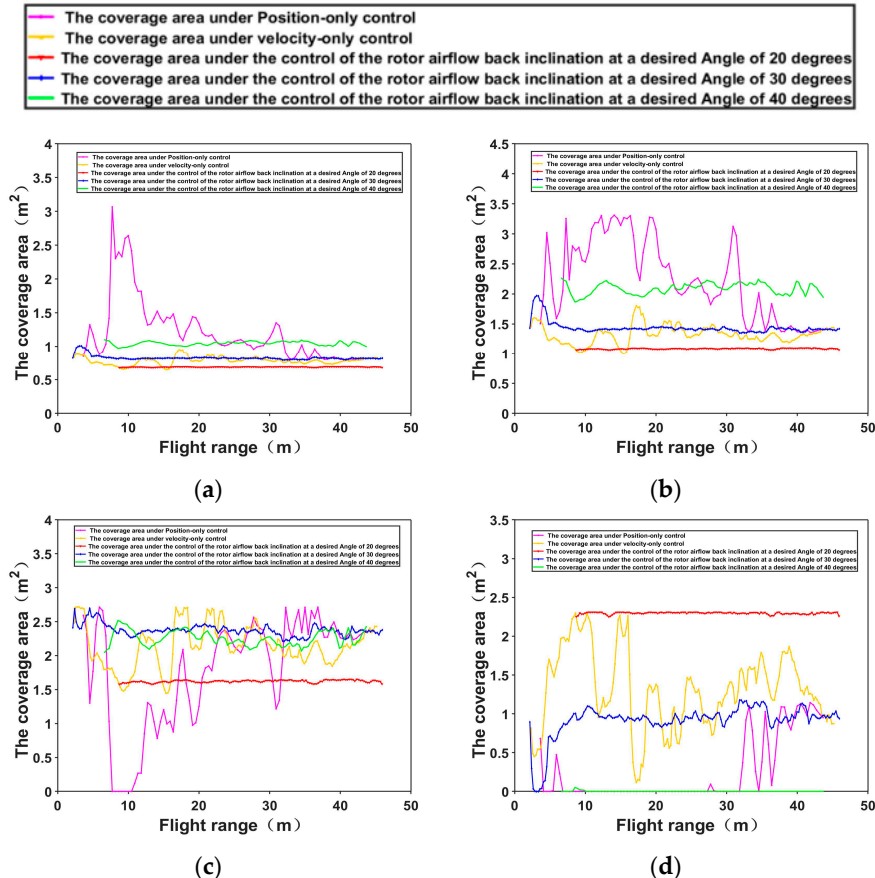

**Figure 13.** Experimental results of the coverage area of a rotary-wing UAV at different control modes: (**a**) $H_{UAV\_to\_Canopy} = 1$; (**b**) $H_{UAV\_to\_Canopy} = 2$; (**c**) $H_{UAV\_to\_Canopy} = 3$; (**d**) $H_{UAV\_to\_Canopy} = 4$.

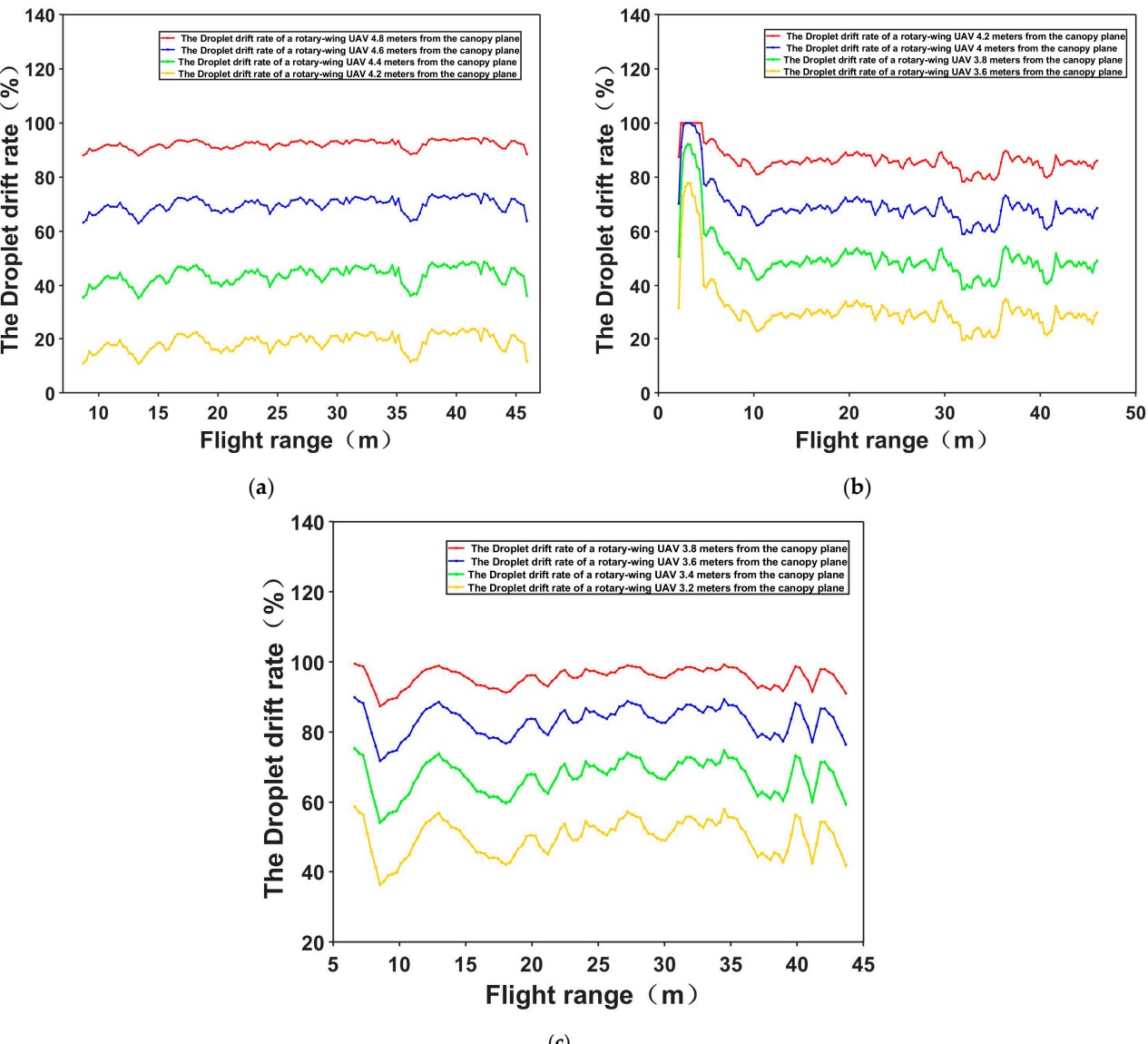

**Figure 14.** Experimental results of the droplet drift rate of a rotary-wing UAV at different distances from the canopy plane: (**a**) expected angle = 20°; (**b**) expected angle = 30°; (**c**) expected angle = 40°.

### 3.3. Comparison and Analysis of the Control Effect and Operation Effect between the Traditional Flight Controller and RABTA Controller

To compare the control effects of the traditional flight controller and RABTA controller and their influence on the field operation effect, backward tilt angle data and the corresponding coverage area under position control, speed control, and backward tilt angle control with the expected angle of 30 degrees in Section 3.2 are extracted for comparative analysis, and the comparison results are shown in Table 7.

As shown in Table 7, the expected angle of the backward tilt angle under the three control modes is 30 degrees. Taking the expected angle as the standard, the steady-state error of the backward tilt angle controlled by the RABTA controller in the actual flight is 0.92 degrees, which is much smaller than the steady-state error of the position control of 10.96 degrees and the steady-state error of the speed control of 7.84 degrees. The independent sample *t*-test is further carried out on the state of the backward tilt angle under the three control modes in the actual flight. The results show that the *p*-value of the backward tilt angle under the control of the RABTA controller is 0.0837, which is greater than 0.05, proving that there is no statistically significant difference between the true angle and the

expected angle of the backward tilt angle. In the other two control modes, the *p*-values of the backward tilt angle are $4.8865 \times 10^{-20}$ and $2.2761 \times 10^{-17}$, respectively, which are both less than 0.01. Therefore, there is a statistically significant difference between the true angle and the expected angle of the backward tilt angle, indicating that the RABTA controller can accurately control the backward tilt angle in the actual operation process, but the traditional UAV flight controller cannot accurately control the state of the backward tilt angle.

**Table 7.** The comparison and analysis table of the backward tilt angle data and the corresponding coverage area under position control, speed control, and backward tilt angle control with the expected angle of 30 degrees.

| Control Mode | Backward Tilt Angle (Degree) | | | One-Sample *t*-Test | | Vertical Distance of the UAV from the Crop Canopy Level (m) | Coverage Area (m$^2$) | | |
|---|---|---|---|---|---|---|---|---|---|
| | Expected Value | Mean Value | Steady State Error | *t* Value | *p*-Value | | Mean Value | Standard Deviation | Variable Coefficient |
| Position | 30 | 40.96 | 10.96 | 11.9076 | $4.8865 \times 10^{-20}$ | 1 | 1.21 | 0.47 | 0.39 |
| | | | | | | 2 | 2.24 | 0.66 | 0.30 |
| | | | | | | 3 | 1.78 | 0.81 | 0.46 |
| | | | | | | 4 | 0.25 | 0.42 | 1.63 |
| Velocity | 30 | 27.84 | 7.84 | −9.3934 | $2.2761 \times 10^{-17}$ | 1 | 0.78 | 0.08 | 0.10 |
| | | | | | | 2 | 1.32 | 0.20 | 0.15 |
| | | | | | | 3 | 2.12 | 0.40 | 0.19 |
| | | | | | | 4 | 1.35 | 0.65 | 0.48 |
| Rotor airflow back tilt angle | 30 | 30.92 | 0.92 | 1.7389 | 0.0837 | 1 | 0.83 | 0.05 | 0.06 |
| | | | | | | 2 | 1.47 | 0.14 | 0.10 |
| | | | | | | 3 | 2.41 | 0.09 | 0.04 |
| | | | | | | 4 | 0.82 | 0.28 | 0.34 |

In addition, when the vertical height ($H_{UAV\_to\_Canopy}$) of the rotor wing UAV from the crop canopy remained constant within the range of 1 m to 4 m, the coverage area (*S1*) of the rotor airflow on the crop canopy was only related to the backward tilt angle. However, the fluctuation of the backward tilt angle under the control of the RABTA controller is only 0.92 degrees. At this time, the variation in the vertex X coordinate ($CX_{max}$) closest to the UAV and the vertex X coordinate ($CX_{min}$) furthest away from the UAV of the rotor airflow coverage on the crop canopy is very small. According to the coverage area calculation formula of Equation (14), the variation coefficients of the cover area of the rotor airflow on the crop canopy are 0.06 (1 m), 0.1 (2 m), 0.04 (3 m), and 0.34 (4 m). All are less than the variation coefficients of the rotor airflow coverage under position control (0.39, 0.3, 0.46, and 1.63) and velocity control (0.1, 0.15, 0.19, and 0.48) at the same UAV operating flight altitude. The results show that the rotor airflow coverage area under the control of the RABTA controller is smaller and more stable than that under the control of the traditional UAV flight controller when the operating flight height of the rotor UAV is unchanged.

In summary, compared with the traditional UAV flight controller, the RABTA controller can take the backward tilt angle as the control object to realize the effective control of the backward tilt angle during the UAV field application operation, thereby reducing the fluctuation of the rotor airflow coverage area and improving the distribution uniformity of pesticide droplets in the crop canopy during the UAV application process. It improves the effect of the UAV field application.

### 3.4. Comparison and Analysis of the Control Effects and Operation Effects of the RABTA Controller under Different Expected Angles

To compare the control effect of the RABTA controller under different expected angles and its influence on the field operation effect, the angle data of the backward tilt angle under the RABTA controller with the expected angles of 20 degrees, 30 degrees, and 40 degrees in Section 3.2, the corresponding coverage area of the rotor airflow in the crop canopy, and the drift rate of fog droplets under the different flight heights of the UAV are extracted for comparative analysis, and the comparison results are shown in Table 8.

**Table 8.** The comparison and analysis table of the angle data of the backward tilt angle under the RABTA controller with the expected angles of 20 degrees, 30 degrees, and 40 degrees, the corresponding coverage area of the rotor airflow in the crop canopy, and the drift rate of fog droplets under the different flight heights of the UAV.

| Expected Value (Degree) | Steady Rotor Flow Back Angle (Degree) | | Adjust Time (s) | One-Sample *t*-Test | | Vertical Distance of the UAV from the Crop Canopy Level (m) | Average Coverage Area (S1) (m²) | Vertical Distance of the UAV from the Crop Canopy Level (m) | Average Drift Rate (τ) (%) |
|---|---|---|---|---|---|---|---|---|---|
| | Mean Value | Error | | *t* Value | *p*-Value | | | | |
| 20 | 20.27 | 0.27 | 3.3 | 1.1441 | 0.2545 | 1 | 0.68 | 4.8 | 91.77 |
| | | | | | | 2 | 1.08 | 4.6 | 69.27 |
| | | | | | | 3 | 1.62 | 4.4 | 42.98 |
| | | | | | | 4 | 2.30 | 4.2 | 17.98 |
| 30 | 30.92 | 0.92 | 2.7 | 1.7389 | 0.0837 | 1 | 0.83 | 4.2 | 88.19 |
| | | | | | | 2 | 1.47 | 4 | 72.47 |
| | | | | | | 3 | 2.41 | 3.8 | 54 |
| | | | | | | 4 | 0.82 | 3.6 | 35.02 |
| 40 | 40.05 | 0.05 | 2.9 | 0.6135 | 0.5408 | 1 | 1.05 | 3.8 | 95.73 |
| | | | | | | 2 | 2.09 | 3.6 | 83.32 |
| | | | | | | 3 | 2.25 | 3.4 | 67.44 |
| | | | | | | 4 | 0.002 | 3.2 | 50.12 |

It can be seen from Table 8 that according to the 5% error band proposed by the automatic control principle, the steady-state average values of the backward tilt angle under three different expected angles are all within the expected error band, the maximum steady-state error is ±0.92 degrees, and the maximum adjustment time is only 3.3 s. Furthermore, an independent sample t-test is carried out for three different expected angles and their corresponding backward tilt angles. The results show that the *p*-values of the backward tilt angle corresponding to the expected angles of 20 degrees, 30 degrees, and 40 degrees are 0.2545, 0.0837, and 0.5408, respectively, which are all greater than 0.05. Therefore, there is no significant difference between the true angle and the expected angle of the backward tilt angle. The above data show that in the process of UAV field operations, the RABTA controller can accurately and quickly control the state of the backward tilt angle while taking into account the operation task.

In addition, when the vertical height ($H_{UAV\_to\_Canopy}$) of the rotor UAV from the crop canopy changes from 4 m to 3 m and the backward tilt angle remains 20 degrees, according to Equations (15) and (16), the difference between the vertex X coordinate ($CX_{max}$) closest to the UAV and the vertex X coordinate ($CX_{min}$) farthest away from the UAV of the coverage of the rotor airflow on the crop canopy decreases. At this time, the coverage area $S_1$ of the rotor airflow in the crop canopy changed from 2.3 m² to 1.62 m², and the fluctuation was as high as 0.68 m². To reduce the fluctuation of the coverage area, as shown in Table 8, the RABTA controller can be used to control the backward tilt angle to 30 degrees to change the vertex X coordinate ($CX_{max}$) closest to the UAV and the vertex X coordinate ($CX_{min}$) farthest away from the UAV of the rotor airflow coverage on the crop canopy by changing the starting point X coordinate ($CX_{start}$) of the rotor airflow axis. At this time, the coverage area of the rotor airflow in the crop canopy is adjusted to 2.41 m², which reduces the fluctuation of the coverage area. The above analysis shows that in the process of field applications, the RABTA controller can control the backward tilt angle in real time to maintain a better state to effectively control the coverage area of the rotor airflow in the crop canopy, reduce the variation and fluctuation of the coverage area, improve the distribution uniformity of pesticide droplets in the crop canopy, and improve the effect of field applications.

When the vertical height ($H_{UAV\_to\_Canopy}$) of the rotor-wing UAV from the crop canopy changes from 3.2 m to 3.6 m and the backward tilt angle remains at 40 degrees, according to Equation (16), the vertex space Z coordinate of the intersection between the rotor airflow cutoff section and the crop canopy level (*RZ*) will decrease. As a result, the X-coordinate of the vertex space of the intersection between the rotor airflow cutoff section and the crop

crown level (*RX*) increased, and the y-coordinate of the vertex space of the intersection between the rotor airflow cutoff section and the crop crown level (*RY*) decreased so that the drift rate of pesticide droplets increased from 50.12% to 83.32%. To reduce the drift rate of pesticide droplets, as shown in Table 8, the backward tilt angle can be controlled to 30 degrees by the RABTA controller to increase *RZ*. At this time the *RX* is reduced, the *RY* is increased, and the drift rate of pesticide droplets is reduced to 35.02%. The above analysis shows that in the field application process of the rotor UAV, the RABTA controller can control the backward tilt angle in real time to maintain a better state, thereby reducing the fog drop drift rate in the process of application, reducing the pesticide fog drop drift, and improving the effect of field application.

In summary, in the actual field application process of UAVs, the RABTA controller can accurately and quickly control the backward tilt angle to maintain a better state to complete the operation when the operation height changes to effectively control the rotor air coverage area in the crop canopy to maintain a stable state and reduce the fog drift rate. Moreover, the distribution uniformity of pesticide droplets in the crop canopy is improved, the drift of fog droplets is reduced, and the application effect of UAVs in the field is improved.

## 4. Conclusions

We aimed to solve the problem that because the UAV flight controller does not take the backward tilt angle as the control object in the process of field application, it is unable to control the backward tilt angle accurately and quickly, which is a measure of the operation effect in the process of application. In this paper, the RABTA controller is established, and the control effect and operation effect of the controller are experimentally analyzed in an actual flight environment. The conclusions are as follows:

1.  In the field application process of the rotor UAV, the RABTA controller can replace the traditional UAV flight controller to control the backward tilt angle and maintain the control error below 1 degree, achieving a 48.3% improvement in the uniformity of the distribution of pesticides droplets across the crop canopy. This will provide a new operational control mode for agricultural UAV, which is a new direction for efficient and high-quality operations with field work effect factors as the control objects, achieving innovation in the control paradigm of agricultural UAV.
2.  In the actual field application process of the rotor UAV, the RABTA controller can accurately and quickly control the backward tilt angle in the range of 20 degrees to 40 degrees to maintain a better state to complete the work (the maximum steady-state error is 0.92 degrees, and the maximum adjustment time is 3.3 s), reducing the degree of the non-uniform distribution of droplets by 83.8% and the drift rate of droplets by 48.3%. On the basis of retaining the high efficiency of agricultural drone operations, it introduces new indicators for enhanced effectiveness, increases crop yield, and achieves economic benefits in agriculture.

**Author Contributions:** H.W.: Conceptualization, Methodology, Writing—original draft, Formal Analysis, Software, Visualization, Data Collection, Data curation, Investigation, Validation, Writing—review and editing; D.L.: Investigation, Validation, Methodology, Data Collection; Y.Z.: Investigation, Validation, Methodology, Data Collection; Z.L. (Zongru Liu): Investigation, Validation, Data Collection; Y.L.: Investigation, Software, Data Collection; Z.L. (Zhijie Liu): Investigation, Software, Investigation, Data Collection; T.H.: Investigation, Validation, Data Collection; K.L.: Software, Data Collection; S.X.: Resources, Data Collection, Supervision; J.L.: Conceptualization, Resources, Writing—review and editing, Funding acquisition, Supervision, Project administration. All authors have read and agreed to the published version of the manuscript.

**Funding:** This work was supported in part by the Guangdong Basic and Applied Basic Research Foundation under Grant 2023A1515011932 and in part by the Key Technologies Research and Development Program of Guangzhou under Grant 202206010164 and Grant 2023B03J1323.

**Data Availability Statement:** The raw/processed data required to reproduce the above findings cannot be shared at this time, as the data also forms part of an ongoing study.

**Conflicts of Interest:** The authors declare that they have no known competing financial interests or personal relationships that could have appeared to influence the work reported in this paper.

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
