# Peer review of "Establishment and Verification of the UAV Coupled Rotor Airflow Backward Tilt Angle Controller"

_drones, doi:10.3390/drones8040146_

Round 1
Reviewer 1 Report
Comments and Suggestions for Authors
Review of “Establishment and verification of the UAV coupled rotor air- flow backward tilt angle controller”
In this paper entitled “Establishment and verification of the UAV coupled rotor air- 2 flow backward tilt angle controller”, the authors introduced a rotor airflow backward tilt angle controller (RABTA) to control the backward tilt angle during operation
Recommendation: This reviewer recommends publishing this paper after some minor modifications. The paper shows a deep understanding for the underpinning physics and the physical interpretation of the results presented. The manuscript will be an added value to the community
Minor concerns:
1- Table 1 , replace \ by NA
2- Table 2, replace \ by NA
3- Table 3, can you elaborate the uncertainty analysis based on the error presented in this table and the precision of the angle measurement.
4- Table 6, represents the truncation error for velocities and angles up to six digits which is not logic for the precision sensors that used, please elaborate.
5- Line 414, rephrase the cation to Experimental results for rear inclination with flight range at different designed angles.
6- Table 7and table 8, truncated the values presented in this table.
Author Response
Thank you very much for taking the time to review this manuscript. We have supplemented and corrected the errors in table formatting, image cation, and logical errors in the article based on your suggestions. Thank you once again for reviewing this manuscript.

Reviewer 2 Report
Comments and Suggestions for Authors
My comments and suggestions:
Abstract:
- The abstract can benefit from a brief mention of the methodology and the technology used in the RABTA sensor which can give a clear picture of the innovation. A sentence like "The RABTA controller integrates advanced sensor technology with a novel algorithmic approach, utilizing real-time data acquisition and state-space analysis to dynamically adjust the UAV’s rotor airflow, ensuring precise control of the backward tilt angle."
- While the abstract highlighted the main findings, including a specific statistical results or a quantitative improvements would definitely enhance its impact of the reader. Something like: "A comparative analysis demonstrated that the RABTA controller reduces the error to less than 1 degree, achieving a xx% improvement in the uniformity of the distribution of pesticides droplets across crop canopy". This addition would provide a concrete measure of the RABTA's effectiveness and highlights the precise control improvement. This parts can be also added to the conclusion section as it lacks this quantitative elaboration.
Design of the RABTA sensor
- In this section you have mentioned the design considerations for the tail plate's transverse width and its position relative to the rotorcraft. I would like to see how do these design elements contribute to the sensor's ability to accurately measure the backward tilt angle? And if there is comparative data or theoretical backing that supports these specific dimensions and configurations?
Experimental and Result Analysis
- The experiment is described as taking place in a windless environment, which helps control variables. I would like to see a discussion on how this controlled setting compares to other real-world conditions/scenarios where wind and other factors might affect the sensor’s performance. And how such conditions would limit its applicability and usability in other scenarios.
- The analysis lacks a comparative perspective with existing sensors or controllers used in UAV applications. Highlighting how the RABTA sensor's performance metrics stack up against current technologies could provide a clearer view of its advancements or limitations.
- A detailed comparison with existing solutions, including other advanced control systems, could further highlight the RABTA controller's unique benefits and innovations.
- In figures such as Fig 12, 13, and 14, please ensure that data plot lines are distinct from one another. You need to vary the line styles or markers. This will help in distinguishing between the different data sets, especially when the figures are reproduced in black and white.
Conclusions
- The conclusions address the immediate findings, they need to be expanded to discuss the broader implications for the field of UAV technology and agricultural applications such as how this research findings might influence future developments in UAV flight control systems for Ag applications?
- The conclusions could touch on the economic and operational benefits of implementing the RABTA controller in real-world UAV applications.
Author Response
Thank you very much for taking the time to review this manuscript. We have supplemented and corrected the errors in abstract, image error, and conclusion in the article based on your suggestions. Thank you once again for reviewing this manuscript.

Reviewer 3 Report
Comments and Suggestions for Authors
See file attached.

Author Response
Thank you very much for taking the time to review this manuscript. We have supplemented and corrected the errors in the details throughout the manuscript based on your suggestions. Thank you once again for reviewing this manuscript.
